# Malaria Transmission and Spillover across the Peru–Ecuador Border: A Spatiotemporal Analysis

**DOI:** 10.3390/ijerph17207434

**Published:** 2020-10-13

**Authors:** Annika K. Gunderson, Rani E. Kumar, Cristina Recalde-Coronel, Luis E. Vasco, Andree Valle-Campos, Carlos F. Mena, Benjamin F. Zaitchik, Andres G. Lescano, William K. Pan, Mark M. Janko

**Affiliations:** 1Duke Global Health Institute, Duke University, Durham, NC 27710, USA; annika.gunderson@duke.edu (A.K.G.); william.pan@duke.edu (W.K.P.); 2Nicholas School of the Environment, Duke University, Durham, NC 27710, USA; rani.kumar@duke.edu; 3Department of Earth and Planetary Sciences, Johns Hopkins University, 327 Olin Hall, 3400 N. Charles Street, Baltimore, MD 21218, USA; grecald1@jhu.edu (C.R.-C.); zaitchik@jhu.edu (B.F.Z.); 4Facultad de Ingeniería Marítima y Ciencias del Mar, Escuela Superior Politécnica del Litoral, Guayaquil 090150, Ecuador; 5Instituto de Geografía, Colegio de Ciencias Biológicas y Ambientales, Universidad San Francisco de Quito, Quito 170104, Ecuador; luisesteban_vasco@hotmail.com (L.E.V.); cmena@usfq.edu.ec (C.F.M.); 6Emerge, Emerging Diseases and Climate Change Research Unit, Universidad Cayetano Peruana Heredia, San Martín de Porres 15102, Peru; avallecam@gmail.com (A.V.-C.); willy.lescano@upch.pe (A.G.L.); 7Institute for Health Metrics and Evaluation, University of Washington, Seattle, WA 98121, USA

**Keywords:** malaria, human mobility, spillover, spatiotemporal modeling, Bayesian methods

## Abstract

Border regions have been implicated as important hot spots of malaria transmission, particularly in Latin America, where free movement rights mean that residents can cross borders using just a national ID. Additionally, rural livelihoods largely depend on short-term migrants traveling across borders via the Amazon’s river networks to work in extractive industries, such as logging. As a result, there is likely considerable spillover across country borders, particularly along the border between Peru and Ecuador. This border region exhibits a steep gradient of transmission intensity, with Peru having a much higher incidence of malaria than Ecuador. In this paper, we integrate 13 years of weekly malaria surveillance data collected at the district level in Peru and the canton level in Ecuador, and leverage hierarchical Bayesian spatiotemporal regression models to identify the degree to which malaria transmission in Ecuador is influenced by transmission in Peru. We find that increased case incidence in Peruvian districts that border the Ecuadorian Amazon is associated with increased incidence in Ecuador. Our results highlight the importance of coordinated malaria control across borders.

## 1. Introduction

Malaria remains a major health threat globally, with 228 million cases leading to 405,000 deaths worldwide in 2018 [1]. While staggering, these numbers represent declines in transmission and mortality since 2010, when there were an estimated 251 million cases and 585,000 deaths [1]. The burden of malaria is unevenly distributed around the world, with the WHO’s African region accounting for 93% (213 million) of the cases and 94% of the deaths reported in 2018 [1]. Furthermore, the incidence rate of *Plasmodium falciparum*, the parasite species contributing the greatest to mortality, is estimated to be around 500 cases per thousand people per year in sub-Saharan Africa, compared to the 250 cases per thousand people per year in the Americas and Central and South Asia combined; however, *Plasmodium vivax* is the main parasite causing malaria in South America, contributing to 70–80% of cases [1,2,3].

In addition to an uneven burden of malaria worldwide, progress in reducing transmission has been uneven as well. For example, while Africa has seen dramatic declines in transmission between 2010 and 2018, Central and South America saw a 14% increase (1.47% annually) in cases, as well as a 26% increase in deaths, during this same period. Four countries, Peru (6%), Colombia (10%), Brazil (23%), and Venezuela (51%), contributed 90% of the cases of malaria in South America in 2018. Furthermore, the Americas region is one of only two regions (along with the Eastern Mediterranean region) that saw an increase in cases [1].

A number of factors contribute to the rising incidence rate in the Americas. Perhaps the most important factor has been the lack of adequate funding for malaria control. For example, 84% of funding for malaria control originates from domestic sources within South America, and overall funding decreased by 23% from 2010 to 2018, following a period in which supplemental support for malaria control from the Global Fund led to dramatic declines in transmission [1]. In addition to disparities in resource allocation for malaria control, human movement has also been linked to rising rates of malaria transmission in places where it had been under control, including in South America [4]. For example, malaria cases in Venezuela accounted for half those in the Americas, a 365% increase from 2000 to 2015, and mass migration from the country due to the deteriorating social and economic situation is compounding malaria risk in neighboring countries. Colombia and Brazil have seen large percentages of malaria cases imported from Venezuela (78% and 81%, respectively), overwhelming healthcare facilities [5]. Furthermore, many of the migrants from Venezuela are moving south through Colombia, Ecuador, and Peru to the Southern Cone, leading to imported and autochthonous cases of malaria in the Ecuador–Peru border region, an area that has twice eliminated malaria [6]. This migration is particularly important for Ecuador, which is 1 of 21 countries targeted for malaria elimination in 2020 [7,8].

Much of the Ecuador–Peru border region is located within the broader Amazon basin, which is home to a largely rural population living in communities located along the river systems that serve as tributaries for the Amazon river, which begins in Loreto, Peru and flows eastward through Brazil and into the Atlantic ocean. Those living on both sides of the Ecuador–Loreto border work in a variety of industries, including agriculture, fishing, timber, and oil and mining extraction. Those who work in extractive sectors often need to travel long distances along the river network to reach forest concessions or oil fields [2,4,9,10]. Additionally, indigenous communities share family, kinship, and commercial ties across borders, each of which promotes mobility across borders. Such travel poses a risk for malaria-eliminating countries such as Ecuador, since intermittent travel across borders has been associated with malaria diffusion [8,9,10,11,12,13,14]. However, work in this area is limited, and tends to focus on a small number of locations and time points. Therefore, because malaria is increasing near the country’s border with the Loreto region of Peru, this paper aims to understand the relationship between cross-border malaria transmission and the role of human movement along the river systems that span the Ecuador–Peru border.

## 2. Materials and Methods

### 2.1. Study Population

Our primary outcomes were the number of microscopy-confirmed *P. falciparum* and *P. vivax* cases in each canton in the Ecuadorian Amazon during each epidemiological week from 2006 to 2018. In Ecuador, the National Malaria Eradication System (SNEM) and the Ministry of Health collect and report individual cases of malaria across the country collected from local health posts. These reports are compiled within their respective cantons before being sent to the Ministry of Health. Given that our interest was in malaria across borders, we further compiled case counts for *P. vivax* and *P. falciparum* from districts in Loreto, which uses the same reporting system as Ecuador.

### 2.2. Primary Exposures

To assess the degree to which cross-border malaria transmission was driven by connectivity to the river networks connecting the Ecuadorian Amazon to the Peruvian Amazon, we considered two primary exposures. First, to assess the degree to which river connectivity across the Ecuador–Loreto border was associated with malaria, we constructed binary indicators for whether the Putumayo, Napo, Curaray, Tigre, and Pastaza rivers pass through a canton (Ecuador) or district (Loreto). We chose these rivers because they represent the major cross-border transportation corridors among those living in the Ecuadorian Amazon. Second, to assess the degree to which malaria transmission in the Ecuadorian Amazon was associated with malaria transmission across the border in Loreto, we constructed a lagged exposure variable for both *P. falciparum* and *P. vivax.* Specifically, for each canton, we calculated the cumulative incidence during the prior month in the Loreto district that was connected to the canton by one of the river corridors noted above. Figure 1 shows a map of the study area, including the river network connecting the Ecuadorian Amazon and Loreto, Peru.

### 2.3. Environmental Confounders

Malaria transmission is an environmentally driven disease. For example, abundance of the *Anopheles* mosquito populations responsible for transmission fluctuates in relation to rainfall, water levels, and temperature [15,16]. Additionally, travel along the river networks is likely dependent on environmental factors such as rainfall. For these reasons, we included a number of environmental confounders. Specifically, we included canton-level measures of weekly average rainfall (mm), temperature (°C), soil temperature (°C), and soil moisture (kg/m^2^) that were extracted and merged onto the malaria surveillance data. These environmental covariates were generated from a Land Data Assimilation System (LDAS) constructed for the Amazon basin using methods previously described [17,18].

### 2.4. Statistical Methods

We used Bayesian spatiotemporal Poisson regression models to assess the relationship between cross-border connectivity and malaria transmission in the Ecuadorian Amazon. We fit several models designed to assess different assumptions regarding the underlying data-generating process. Given our first hypothesis that river networks connecting Loreto to the Ecuadorian Amazon were drivers of malaria diffusion, we first fit a model with the binary exposures indicating whether Ecuadorian cantons or Loreto districts were connected to the Putumayo, Napo, Curaray, Tigre, or Pastaza rivers, which are the major travel corridors in the region. We compared this model to the same model without the river exposures to assess whether incorporating connectivity led to a better model fit. Second, we subsetted the data to include only the Ecuadorian Amazon cantons, and fit a model with exposure to cross-border malaria transmission in Loreto districts to assess whether increased malaria transmission in Loreto was associated with increased malaria transmission in Ecuador.

We fit separate models for *P. vivax* and *P. falciparum* malaria. All models incorporated independent spatial and temporal random effects. Spatial random effects were modeled using an improper conditionally autoregressive prior (CAR) [19]. Temporal random effects were modeled via an AR1 process. Precision parameters for both spatial and temporal random effects were assigned penalized complexity priors to regularize the model [20]. Regression coefficients were assigned standard normal prior distributions. Model fitting was done using integrated nested Laplace approximation using R-INLA [21]. Data management was done in R (R Foundation for Statistical Computing, Vienna, Austria), Stata (StataCorp LP., College Station, TX, USA), AMOS-23 (IBM, Armonk, NY, USA), and ArcGIS (Esri, Redlands, CA, USA).

## 3. Results

Between 2006 and 2018, there were 9230 cases of *P. vivax* and 499 cases of *P. falciparum* reported across Ecuador. There was very little transmission of malaria of either species during the first two years of the study period. Beginning in 2008, however, *P. vivax* incidence rose dramatically and hovered between 1–4 cases/1000 people/week through 2009, after which transmission began a three-year period of decline and reached a steady state of transmission of under 1/2 cases/1000/week in late 2011. *P. vivax* transmission remained at this low level until May 2014, when it began to rise for the duration of the study period. *P. falciparum* malaria, conversely, remained at low levels across the Ecuadorian Amazon until January 2016, when incidence suddenly spiked to nearly 4 cases/1000/week before falling back down to less than ½ cases/1000/week for the remainder of the study period. Figure 2 shows the incidence over time for both *P. vivax* and *P. falciparum* malaria across the study region, as well as for the cantons/districts making up the Ecuador–Loreto border. As can be seen, *P. vivax* exhibited persistently higher transmission than *P. falciparum* not only over the whole study area, but also in border areas. Additionally, *P. vivax* incidence appeared to rise first in Loreto, Peru, with a subsequent rise in Ecuador. Conversely, there was no such clear pattern for *P. falciparum*, possibly because *P. falciparum* was less common in the region, as well as the fact that Ecuador had eliminated it until relatively recently.

In addition to varying over time, incidence of both *P. vivax* and *P. falciparum* varied geographically. For example, much of the region reported 0 malaria cases over the study period, while Aguarico (number 18 in Figure 1), a canton connected to Loreto via the Napo river, had incidence rates ranging from 0 cases/1000/week to 7.4 cases/1000/week, a peak that was reached during the last week of March 2018. The highest incidence rates of *P. falciparum* also occurred in Aguarico, peaking at 2.3 cases/1000/week in the last week of February 2017. Moreover, cantons bordering the Loreto region of Peru reported considerably more cases of malaria than cantons that did not share a border, with border cantons reporting 61% and 90% of the total *P. vivax* and *P. falciparum* cases, respectively, despite having only 16% of the total population. Figure 3 maps annual incidence rates of both *P. vivax* and *P. falciparum* malaria over the study period.

The results from statistical modeling of connectivity via the cross-border river network indicated that models incorporating the river network outperformed models that did not include the river network. Table 1 shows the out-of-sample predictive performance for models for both *P. vivax* and *P. falciparum*, which shows that the models that included connectivity to the river network had lower root mean square prediction error (RMSPE), indicative of improved model fit, for both species of malaria. Given this improved performance, we turned to estimating the effect of malaria transmission in Loreto districts bordering the Ecuadorian Amazon on malaria transmission in Ecuador. Results from this analysis indicated that increased malaria incidence during the previous month in Loreto districts bordering cantons in Ecuador was associated with increased malaria transmission in those cantons. For example, for *P. vivax*, an increase of 1 case/1000/month in Loreto districts during the prior month was associated with a 3.1% increase of malaria incidence across the border in Ecuador (Rate Ratio [RR] 1.031; 95% uncertainty interval (UI) 1.029–1.033). Results were similar for *P. falciparum*, with a 1-unit increase in incidence in Loreto districts associated with a 3.0% increase in the incidence rate (RR 1.030; 95% UI 1.020–1.039). Beyond these main findings, we also observed that rainfall had a much stronger effect on *P. falciparum* incidence than *P. vivax*, that higher soil temperatures were associated with decreased incidence of *P. vivax*, and that the effects of soil moisture were similar for both *P. vivax* and *P. falciparum*. Table 2 shows results from the regression models.

## 4. Discussion

Malaria transmission in the Amazon varies considerably across space, over time, and by species. This variability is driven by a complex ecology that involves interactions among human and mosquito populations in the environment, all of which are highly diverse, particularly in the Amazon. Our work here sought to address one aspect of that complex ecology—cross-border human movement—and to do so over regional spatial scales and spanning 2006–2018, a period when there were both widespread declines in transmission as well as subsequent resurgences. Doing so presented a number of challenges, including in particular the lack of data on human movement itself. To address this limitation, we constructed two sets of primary exposure variables. First, because the river networks are the primary means of movement in the region, we constructed indicator variables for each of the major tributaries of the Amazon that flow from the Ecuadorian Amazon into Loreto, Peru. Second, we constructed lagged variables of cumulative incidence of both *P. vivax* and *P. falciparum* malaria from the previous month in those Loreto districts connected to cantons in Ecuador via the river network. Our results from modeling river network connectivity suggest that it represented a driver of malaria diffusion in the region, as shown by the improved predictive performance of modeling. Furthermore, our findings from modeling the lagged exposure of malaria incidence among districts immediately across the border from Ecuador indicate that, for each additional case/1000/month, the incidence rate for both *P. vivax* and *P. falciparum* increased by 3%.

Our work here has a number of limitations. First, the measures of connectivity implemented in this study serve as a proxy of human movement, and are therefore too simplistic. Indeed, human movement involves a complex set of drivers ranging from familial ties to economic considerations that we do not account for here [14,22]. Additionally, our malaria data were collected via passive surveillance, which limits the generalizability of our findings, as asymptomatic cases or individuals unable to reach a clinic to be diagnosed are missed in our data. Thus, our malaria data underestimate the true disease burden. We also lack data on interventions that occurred over the course of the 13-year study period since these are not part of the routine surveillance system. Our results may therefore be confounded, since those most likely to travel are simultaneously more likely to become infected with malaria, as well as less likely to be protected by interventions. Finally, the coarse spatial resolution of our study masks the highly localized ecology of human–mosquito interactions. However, malaria control planning is conducted at these coarser scales, and as such our work is consistent with policy-making scales.

Despite these limitations, our work here begins to address the need to characterize the effect of human movement on malaria transmission in a setting of malaria elimination. Our study area is unique in that Ecuador is striving to eliminate malaria, but is currently not on track to succeed [8]. One reason for this may be due to malaria transmission across the border in Loreto, Peru, where malaria has been resurgent. Our results suggest that this resurgence in Peru, combined with connectivity via the Amazon’s major tributaries, is undermining elimination efforts in Ecuador. Elimination efforts in Ecuador, therefore, will require a number of measures, including efforts to target highly mobile groups to limit transmission potential [9,13]. Additionally, these efforts will need to be coordinated, at different spatial and temporal scales, with malaria control programs across the border in Peru, as well as with settings further along the river system. To support this effort, binational control strategies can create an open environment for data sharing and sustainable surveillance systems [6]. Furthermore, such binational surveillance systems will need to include active surveillance. Indeed, because prior work in the Amazon has observed that malaria transmission is sustained by highly mobile populations with asymptomatic infections, a critical population to sustaining malaria transmission in the region is not being adequately captured by routine surveillance, either within a country’s borders or beyond them. As a result, cross-border active surveillance efforts to locate these populations must account for the kinship, ethnic, and commercial ties that exist between peoples in the region. Doing so will help ensure that surveillance is linked to the social and economic processes driving transmission. Accomplishing this will further require the support of local communities and stakeholders in the region.

## Figures and Tables

**Figure 1 ijerph-17-07434-f001:**
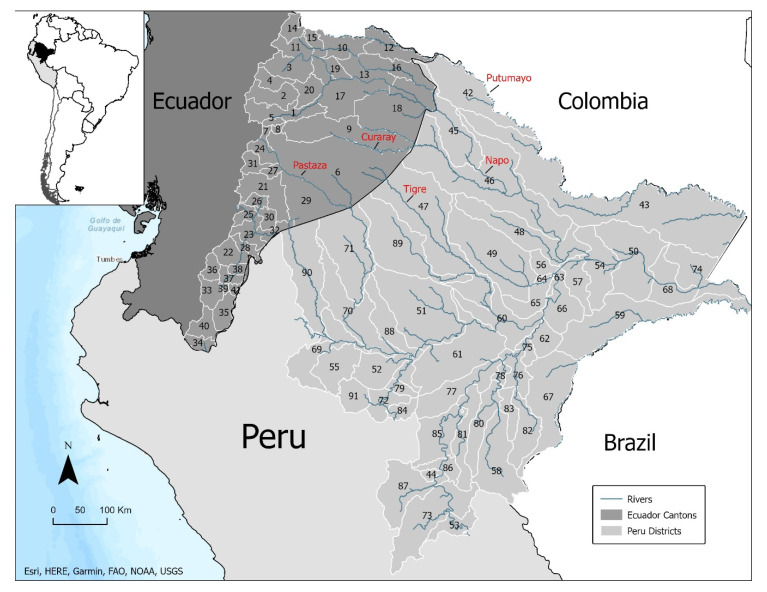
Map of the study area with cantons/districts numbered. 1: Tena, 2: Archidona, 3: El Chaco, 4: Quijos, 5: Carlos Julio Arosemena Tola, 6: Pastaza, 7: Mera, 8: Santa Clara, 9: Arajuno, 10: Lago Agrio, 11: Gonzalo Pizarro, 12: Putumayo, 13: Shushufindi, 14: Sucumbíos, 15: Cáscales, 16: Cuyabeno, 17: Orellana, 18: Aguarico, 19: La Joya De Los Sachas, 20: Loreto, 21: Morona, 22: Gualaquiza, 23: Limón Indanza, 24: Palora, 25: Santiago, 26: Sucua, 27: Huamboya, 28: San Juan Bosco, 29: Taisha, 30: Logrodo, 31: Pablo Sexto, 32: Tiwintza, 33: Zamora, 34: Chinchipe, 35: Nangaritza, 36: Yacuambi, 37: Yantzaza, 38: El Pangui, 39: Centinela Del Condor, 40: Palanda, 41: Paquisha, 42: Teniente Manuel Clavero, 43: Putumayo, 44: Inahuaya, 45: Torres Causana, 46: Napo, 47: Tigre, 48: Mazan, 49: Alto Nanay, 50: Pebas, 51: Urarinas, 52: Jeberos, 53: Padre Marquez, 54: Las Amazonas, 55: Cahuapanas, 56: Punchana, 57: Indiana, 58: Alto Tapiche, 59: Yavari, 60: Nauta, 61: Parinari, 62: Saquena, 63: Belen, 64: Iquitos, 65: San Juan Bautista, 66: Fernando Lores, 67: Yaquerana, 68: San Pablo, 69: Barranca, 70: Pastaza, 71: Andoas, 72: Yurimaguas, 73: Contamana, 74: Ramon Castilla, 75: Jenaro Herrera, 76: Requena, 77: Puinahua, 78: Capelo, 79: Santa Cruz, 80: Emilio San Martin, 81: Maquia, 82: Soplin, 83: Tapiche, 84: Teniente Cesar Lopez Rojas, 85: Sarayacu, 86: Vargas Guerra, 87: Pampa Hermosa, 88: Lagunas, 89: Trompeteros, 90: Morona, 91: Balsapuerto.

**Figure 2 ijerph-17-07434-f002:**
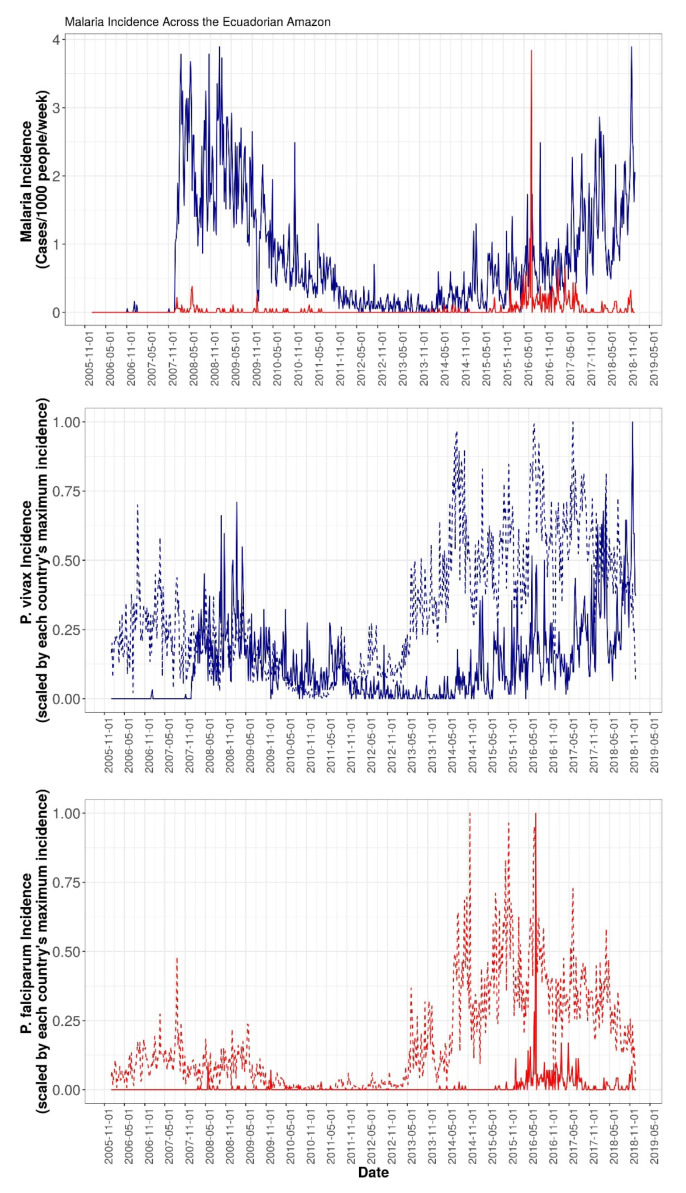
(**Top**) Incidence over time for both *P. falciparum* (red) and *P. vivax* (blue) malaria across the Ecuadorian Amazon from January 2006 to December 2018. (**Middle**) Incidence of *P. vivax* malaria in Loreto districts bordering Ecuador (dotted line) and in Ecuadorian cantons bordering Loreto (solid line). (**Bottom**) Incidence of *P. falciparum* malaria in Loreto districts bordering Ecuador (dotted line) and in Ecuadorian cantons bordering Loreto (solid line).

**Figure 3 ijerph-17-07434-f003:**
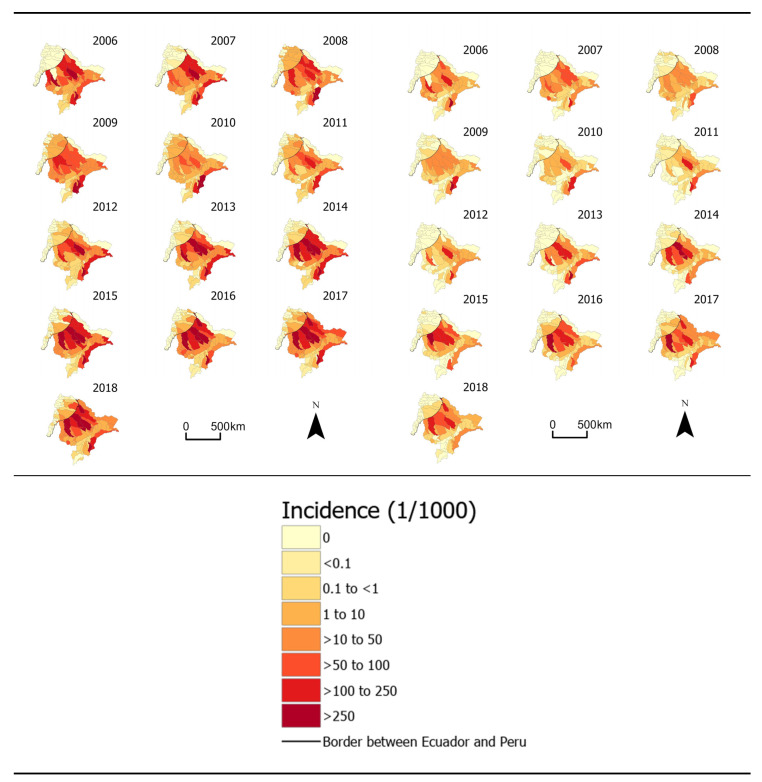
Maps of annual malaria incidence rates for both *P. vivax* (**left**) and *P. falciparum* (**right**) across the study region.

**Table 1 ijerph-17-07434-t001:** Out-of-sample predictive performance based on root mean square prediction error (RMSPE) of models assessing river connectivity and malaria.

Model	*P. vivax*	*P. falciparum*
With river connectivity indicator variables	0.16	0.04
Without river connectivity indicator variables	1.26	0.88

**Table 2 ijerph-17-07434-t002:** Incidence rate ratio estimates from spatiotemporal regression models on the effect of cross-border malaria transmission.

Variable	*P. vivax*			*P. falciparum*		
	Estimate	Lower UI	Upper UI	Estimate	Lower UI	Upper UI
Rainfall (mm)	1.022	0.876	1.191	2.336	1.254	4.322
Temperature (°C)	1.501	1.054	2.137	2.924	0.374	22.01
Soil Temperature (°C)	0.528	0.381	0.730	0.808	0.145	4.624
Soil Moisture	0.926	0.829	1.034	1.027	0.665	1.592
Border incidence	1.031	1.029	1.033	1.030	1.021	1.039

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
