# Peer review of "Malaria Transmission and Spillover across the Peru–Ecuador Border: A Spatiotemporal Analysis"

_ijerph, 2020, doi:10.3390/ijerph17207434_

Round 1

Reviewer 1 Report

Gunderson et al. provide a careful analysis of the influence that malaria transmission in Peru exerts on malaria transmission in Ecuador, particularly at regions where the two countries border each other. Their main conclusion is that case incidence in one country is influenced by that observed in the other, pointing towards the need for the implementation of coordinated malaria control measures across borders.

The manuscript is reasonably well written, although some improvements in terms of grammar and language should be introduced. I suggest that the manuscript be revised to correct a few mistakes in this regard. The Introduction is informative and sets the stage nicely for the matter at hand. The Methods also appear to be well-described and appropriate. The Results are presented in an intelligible fashion, and the conclusions are supported by the data. The Discussion is brief and includes only a few references to previous work, and should be improved in that regard. In addition, I have only a few concerns/remarks for the authors’ consideration:

In the legend of Figure 3, the authors mention P. vivax (top) and P. falciparum (bottom). Do they mean left and right? Please clarify. Also, it would be nice to use the same color coding as in Figure 2, with shades of red for P. falciparum and shades of blue for P. vivax.

An interesting finding is that the statistical modeling of connectivity via the cross-border river network indicate that models incorporating the river network outperformed models that did not include the river network. This is perhaps unsurprising, as the river network likely contributes to incidence, but maybe the authors could explain a little more about the differences between the two models and the meaning of the data in Table 1.

Although the authors candidly acknowledge several limitations of their work, one aspect that is hardly discussed and that, to me, appears relevant in this context is the issue of mosquito mobility. The study focuses entirely on the mobility of human populations across borders, which undoubtedly plays a role, since an infected person may be carrying transmissible forms of the parasites across. However, mosquitoes know no borders and it is unclear to me how their mobility might or might not play a role in this regard. I fully understand that this is not the focus of this article, but a few words about this in the Discussion, perhaps with references to work done by others in this regard, would be welcome.

Finally, the authors finish the Abstract by stating that “Our results highlight the importance of coordinated malaria control across borders”. In my opinion, the Discussion would benefit from the author’s insights or suggestions about what such measures should be and how they should be implemented.

Author Response

To Reviewer 1

Gunderson et al. provide a careful analysis of the influence that malaria transmission in Peru exerts on malaria transmission in Ecuador, particularly at regions where the two countries border each other. Their main conclusion is that case incidence in one country is influenced by that observed in the other, pointing towards the need for the implementation of coordinated malaria control measures across borders.

The manuscript is reasonably well written, although some improvements in terms of grammar and language should be introduced. I suggest that the manuscript be revised to correct a few mistakes in this regard. The Introduction is informative and sets the stage nicely for the matter at hand. The Methods also appear to be well-described and appropriate. The Results are presented in an intelligible fashion, and the conclusions are supported by the data. The Discussion is brief and includes only a few references to previous work, and should be improved in that regard. In addition, I have only a few concerns/remarks for the authors’ consideration:

  • We have reread and edited the manuscript, and made edits where we thought they would improve the paper, as well as according to suggested changes made by other reviewers.
  • We thank the reviewers for noting the introduction, methods, and results are presented in a clear form, and that the conclusions are supported by the data
  • We thank the reviewer for noting the brief discussion. While there is little work on human movement and the spread of malaria in the Amazon, we have expanded our discussion by including an example of what a future malaria control program might consider, namely that control efforts in the region should include active surveillance designed to target highly mobile, asymptomatic carriers of malaria, which prior work has linked to sustained transmission in the region. Specifically, lines 244-54 read: To support this effort, binational control strategies can create an open environment for data sharing and sustainable surveillance systems[6]. Further, such binational surveillance systems will need to include active surveillance. Indeed, because prior work in the Amazon has observed that malaria transmission is sustained by highly mobile populations with asymptomatic infections, a critical population to sustaining malaria transmission in the region is not being adequately captured by routine surveillance, either within a country’s borders or beyond them. As a result, cross-border active surveillance efforts to locate these populations must account for the kinship, ethnic, and commercial ties that exist between peoples in the region. Doing so will help ensure that surveillance is linked to the social and economic processes driving transmission. Accomplishing this will further require the support of local communities and stakeholders in the region.

In the legend of Figure 3, the authors mention P. vivax (top) and P. falciparum (bottom). Do they mean left and right? Please clarify. Also, it would be nice to use the same color coding as in Figure 2, with shades of red for P. falciparum and shades of blue for P. vivax.

  • We thank the reviewer for noticing this error. We now refer to “P. vivax (left)” and “P. falciparum (right)” in the caption for figure 2.
  • We believe that the color scheme should remain as is in Figure 3. While it does represent a departure from the blue/red theme in the time series plots, we believe that a common color scheme across maps is better. Specifically, a common color scale allows for easier comparisons for the different malaria species. Were we two have two color schemes, and two legends, readers would need to map shades of blue to corresponding shades of red to compare vivax and P. falciparum maps. Conversely, we use the two different colors in the time series plots to distinguish species so that they can appear on the same plot, again for ease of comparison.

An interesting finding is that the statistical modeling of connectivity via the cross-border river network indicate that models incorporating the river network outperformed models that did not include the river network. This is perhaps unsurprising, as the river network likely contributes to incidence, but maybe the authors could explain a little more about the differences between the two models and the meaning of the data in Table 1.

We thank the reviewer for seeking clarification. Table 1 shows the out-of-sample predictive performance of two models. The first model (first row of Table 1) includes indicator variables for whether or not a district/canton is connected to a river. The second model (second row of Table 2) does not include these indicator variables. Both models include the same confounders. As can be seen, the first model yields better out-of-sample predictive performance for both P. falciparum and P. vivax malaria.

To make interpretation easier, we have changed the model descriptions in table 1 to read: “With river connectivity indicator variables” and “Without river connectivity indicator variables.” This is more in line with how we describe these models in the statistical methods section.

Additionally, in lines 186-188, we add text to clarify the results, stating: “Table 1 shows the out-of-sample predictive performance for models for both P. vivax and P. falciparum, which shows that the models that included connectivity to the river network had lower root mean square error (RMSPE), indicative of improved model fit, for both species of malaria.”

Although the authors candidly acknowledge several limitations of their work, one aspect that is hardly discussed and that, to me, appears relevant in this context is the issue of mosquito mobility. The study focuses entirely on the mobility of human populations across borders, which undoubtedly plays a role, since an infected person may be carrying transmissible forms of the parasites across. However, mosquitoes know no borders and it is unclear to me how their mobility might or might not play a role in this regard. I fully understand that this is not the focus of this article, but a few words about this in the Discussion, perhaps with references to work done by others in this regard, would be welcome.

We thank the reviewer for this comment. We do not discuss mosquito mobility because it is not a factor in transmission except in very highly localized studies, since the Anopheles mosquitos responsible for malaria transmission typically travel only a few hundred meters (at most a few kilometers) in their lifetime (which is ~2 weeks). Our focus here is on much larger scales.

Finally, the authors finish the Abstract by stating that “Our results highlight the importance of coordinated malaria control across borders”. In my opinion, the Discussion would benefit from the author’s insights or suggestions about what such measures should be and how they should be implemented.

We thank the reviewer for noting this. As noted above, we have added an example of potential future coordinated malaria control: To support this effort, binational control strategies can create an open environment for data sharing and sustainable surveillance systems[6]. Further, such binational surveillance systems will need to include active surveillance. Indeed, because prior work in the Amazon has observed that malaria transmission is sustained by highly mobile populations with asymptomatic infections, a critical population to sustaining malaria transmission in the region is not being adequately captured by routine surveillance, either within a country’s borders or beyond them. As a result, cross-border active surveillance efforts to locate these populations must account for the kinship, ethnic, and commercial ties that exist between peoples in the region. Doing so will help ensure that surveillance is linked to the social and economic processes driving transmission. Accomplishing this will further require the support of local communities and stakeholders in the region.

Reviewer 2 Report

Malaria transmission and spillover across the Peru-Ecuador border: a spatio-temporal analysis. 

The manuscript is well written except in cases of too long sentence in the abstract. The methods and results are well presented. The aim of the paper needs to be revised to reflect the task performed. The environmental component of the analysis is not reflected in the aim.

Author Response

To Reviewer 2

The manuscript is well written except in cases of too long sentence in the abstract. The methods and results are well presented. The aim of the paper needs to be revised to reflect the task performed. The environmental component of the analysis is not reflected in the aim.

-We thank the reviewer for nothing this inconsistency. We have revised the abstract and removed the reference.

-We have further split the following sentence into two:

ORIGINAL: As a result, there is likely considerable spillover across country borders, particularly along the border between Peru and Ecuador, which exhibits a steep gradient of transmission intensity, with Peru having a much higher incidence of malaria than Ecuador.

REVISED: As a result, there is likely considerable spillover across country borders, particularly along the border between Peru and Ecuador. This border region exhibits a steep gradient of transmission intensity, with Peru having a much higher incidence of malaria than Ecuador.

Reviewer 3 Report

This is a manuscript investigating the phenomenon of cross-border malaria ‘spillover’. The data is from the Peru/Ecuador border regions where Peru experiences a high incidence of malaria compared to Ecuador. The timing, possible causes, and correlations of malaria incidence on either side of the border are investigated. The authors benefit from a rich dataset consisting of weekly malaria incidence numbers in ~100 districts spanning the border over the past 13 years. As stated by the authors the major weakness of the manuscript is the lack of true human movement data. Instead, the presence of rivers connecting the two countries is used as a proxy for human movement. The authors are forthright in this being a major limitation which then leads to an over simplified interpretation. Despite this limitation, the data are strong enough to make several interpretations and I feel add to the general body of knowledge. These findigns could be extrapolated to other border regions which are also facing challenges in the era of malaria elimination.

There are a couple of areas where I feel the manuscript could be improved

Line 53: These funding numbers are fascinating. Could you contrast these numbers with comparable ones from Africa?

Figure 1: There could be more interpretation of these graphs. It seems that the falciparum findings are different from the vivax. Vivax Ecuador seems to always follow Peru, whereas falciparum seems to ‘mirror’ each other without one country following the other. Is it worth commnenting?

Table 2: There is a lot more information in this Table than has been discussed in the text. Could the authors also discuss rainfall (affects falciparum but not vivax)? Also temperature – seems to affect vivax but not falciparum. Even though this is not the ‘crux’ of the manuscript – they are interesting data and could also be discussed.

And also a few very small typo/formatting issues.

Line 151: There is an extra ‘from’

Line 152 vs Line 156: The numbering is formatted as “0.5” once and as “ ½” the second time.

Figure 3 Legend: The graphs are divided as right /left not top / bottom as stated

Line 223: “planning” not “panning”

Line 227: There is an extra “the”

Author Response

To Reviewer 3

This is a manuscript investigating the phenomenon of cross-border malaria ‘spillover’. The data is from the Peru/Ecuador border regions where Peru experiences a high incidence of malaria compared to Ecuador. The timing, possible causes, and correlations of malaria incidence on either side of the border are investigated. The authors benefit from a rich dataset consisting of weekly malaria incidence numbers in ~100 districts spanning the border over the past 13 years. As stated by the authors the major weakness of the manuscript is the lack of true human movement data. Instead, the presence of rivers connecting the two countries is used as a proxy for human movement. The authors are forthright in this being a major limitation which then leads to an over simplified interpretation. Despite this limitation, the data are strong enough to make several interpretations and I feel add to the general body of knowledge. These findigns could be extrapolated to other border regions which are also facing challenges in the era of malaria elimination.

There are a couple of areas where I feel the manuscript could be improved

 Line 53: These funding numbers are fascinating. Could you contrast these numbers with comparable ones from Africa?

We agree that the numbers are compelling. Since the focus of this paper is on South America and the epidemic there, we prefer not to include malaria control financing figures for Africa in the manuscript. That said, malaria control in Africa is largely financed by the international donor community. For example, according to the 2019 world malaria report, only 9% of malaria control in West African countries was financed domestically, compared to 20% in Central Africa, and 11% in East and Southern Africa.

Figure 1: There could be more interpretation of these graphs. It seems that the falciparum findings are different from the vivax. Vivax Ecuador seems to always follow Peru, whereas falciparum seems to ‘mirror’ each other without one country following the other. Is it worth commnenting?

 We thank the reviewer for noting this. We have added the following discussion to the figures:

Lines 158-163: As can be seen, P. vivax exhibits persistently higher transmission than P. falciparum both over the whole study area, but also in border areas. Additionally, P. vivax incidence appears to rise first in Loreto, Peru, with a subsequent rise in Ecuador. Conversely, there is no such clear pattern for P. falciparum, possibly because P. falciparum is less common in the region, as well as the fact that Ecuador had eliminated it until relatively recently.

Table 2: There is a lot more information in this Table than has been discussed in the text. Could the authors also discuss rainfall (affects falciparum but not vivax)? Also temperature – seems to affect vivax but not falciparum. Even though this is not the ‘crux’ of the manuscript – they are interesting data and could also be discussed.

We have included a brief discussion of our results:

Lines 196-199: ). “Beyond these main findings, we also observed that rainfall had a much stronger effect on P. falciparum incidence than P. vivax, that higher soil temperatures were associated with decreased incidence of P. vivax, and that the effects of soil moisture were similar for both P. vivax and P. falciparum.”

We avoid discussion of temperature because the uncertainty intervals overlap (and the UI for P. falciparum is quite wide).

 And also a few very small typo/formatting issues.

 Line 151: There is an extra ‘from’

fixed

Line 152 vs Line 156: The numbering is formatted as “0.5” once and as “ ½” the second time.

fixed

Figure 3 Legend: The graphs are divided as right /left not top / bottom as stated

fixed

Line 223: “planning” not “panning”

fixed

Line 227: There is an extra “the”

fixed